# Subjective versus Objective Measure of Physical Activity: A Systematic Review and Meta-Analysis of the Convergent Validity of the Physical Activity Questionnaire for Children (PAQ-C)

**DOI:** 10.3390/ijerph18073413

**Published:** 2021-03-25

**Authors:** Danilo Marasso, Corrado Lupo, Simone Collura, Alberto Rainoldi, Paolo Riccardo Brustio

**Affiliations:** 1School of Exercise and Sport Sciences (SUISM), University of Torino, 10124 Turin, Italy; danilo.marasso@unito.it (D.M.); simone.collura@edu.unito.it (S.C.); 2NeuroMuscularFunction Research Group, School of Exercise and Sport Sciences (SUISM), Department of Medical Sciences, University of Torino, 10143 Turin, Italy; alberto.rainoldi@unito.it (A.R.); paoloriccardo.brustio@unito.it (P.R.B.)

**Keywords:** PA, MVPA, accelerometer, questionnaire, children

## Abstract

This study aimed to highlight the relationship between moderate-to-vigorous physical activity (MVPA) as assessed by accelerometer devices and the Physical Activity Questionnaire for Children (PAQ-C) to estimate the convergent validity of the questionnaire. A systematic review and a meta-analysis were applied by collecting pertinent studies (PubMed, Web of Science, PsycINFO, and SCOPUS) from 1997 until November 2020. The relationship between PAQ-C and MVPA scores was estimated considering correlation coefficients such as the effect size. Fisher’s transformation was used to convert each correlation coefficient into an approximately normal distribution. The pooled correlations between PAQ-C and MVPA scores were measured by r values after converting the Fisher’s z values back into correlation coefficients for presentation. A total of 13 studies were included in the meta-analysis, and a random effects model was adopted. The pooled correlation between PAQ-C and MVPA scores was significant but with a moderate effect size (r = 0.34 [0.29, 0.39], Z = 15.00, *p* < 0.001). No heterogeneity among the studies was observed (I^2^ < 25%). In conclusion, the results highlighted a moderate relationship (around 0.30–0.40) between PAQ-C and accelerometer measurements. These results suggested to concurrently administer both tools to reach a more comprehensive description of children’s PA, in terms of quality and quantity.

## 1. Introduction

International guidelines recommend that children should accumulate at least 60 min of daily moderate-to-vigorous physical activity (MVPA) including sports and leisure activities [1]. Reaching these recommendations is associated with positive health outcomes, such as the reduction of overweight and obesity [1,2] and risk for chronic diseases and cancer in adulthood [3,4,5], and improvement of psychological well-being (e.g., reduction of depression and stress) [6], cognitive function and academic outcomes [1], and quality of life in general [7]. On the contrary, a low level of physical activity (PA) and the increase of sedentary behavior (e.g., screen time) are associated with both short- and long-term negative health consequences [1,8,9]. Despite the benefits of PA, two-thirds of children do not meet the international recommendations [10] and spend a large part of their waking time in sedentary activities [11,12,13]. Public health actions are necessary to counter this trend [1,11,12]. For this purpose, one of the global targets of the World Health Organization for the next years to prevent non-communicable diseases is to reduce the prevalence of insufficient PA by 15% [12,14]. Thus, an accurate assessment of childhood PA behavior is fundamental to accurately identify the effectiveness and the progression of healthcare interventions [15,16].

Data collection to assess and measure PA behavior can involve subjective and self-report (e.g., diaries/logs, questionnaires, or interviews) procedures, as well as objective and direct instruments including motion sensors (e.g., accelerometers, pedometers, and heart rate monitors), physiological marker (e.g., biomarkers), and calorimetry (e.g., direct and indirect) [15,17]. Among self-report instruments, PA questionnaires (PAQ) are practical to administer, relatively inexpensive, acceptable, and allow an insight into the PA of a large-scale population [18]. Nevertheless, PAQ can present some limitations. Socioeconomic and sociodemographic factors and measurement bias including misinterpretation or deliberate changes (e.g., social desirability) and difficulty to recall activities may affect the final score [18,19]. In particular, it is common that children have difficulties in understanding questions and accurately recalling activities [20], leading to an overestimate or underestimate of the overall PA level [15,21]. An alternative approach to a self-reported measure is the objective monitoring of PA using wearable devices, such as accelerometers. Accelerometer technology allows the measurement of accelerations produced by movement and may be considered an effective and feasible instrument to measure PA levels. Nevertheless, even though accelerometer technology objectively estimates the frequency, duration, and intensity of PA, it is time and cost intensive and difficult to administer to a large-scale population [15,17]. Moreover, due to physical constraints, some specific activities (e.g., swimming or bicycle activities) may be difficult to assess [22].

Among the numerous PAQs for children (see [22] for a complete review), none emerges as the best in terms of psychometric characteristics, but the Physical Activity Questionnaire for Older Children (PAQ-C) is one of the most promising [18]. The PAQ-C has been widely used in research and field settings (e.g., school context) to discern general levels of PA over the last seven days in children aged 8–14 years [23], even if it should not be used to assess PA in the summer or holiday periods [24]. The PAQ-C assesses activities related to common sports, leisure activities, games, and physical education classes [18,23]. It consists of ten items, nine of which are used to calculate summary activity scores with a score ranging from 1 to 5 for each item, where a higher score indicates higher levels of activity [23]. Interestingly, the first item is an activity checklist of common activities aimed to aid children to recall with memory cues. The PAQ-C demonstrated acceptable psychometric properties [18,23] with an acceptable-to-good internal consistency, test–retest reliability, and sensitivity to detect gender differences [23,25,26,27,28]. The PAQ-C demonstrated convergence with athletic competence, enjoyment perception, body mass index, cardiorespiratory, and cardiovascular fitness [16,25,26,27,29]. Nevertheless, while some studies showed a moderate correlation with accelerometer scores, and in particular with MVPA [16,20,26,27,28,30,31,32], others reported a low or no correlation [23,33,34,35,36].

The PAQ-C is used in different countries and cultural contexts for research purposes (e.g., Italy [16], Greece [28], Netherlands [29], China [30]). Thus, it is necessary to better understand the convergent validity (e.g., the extent to which different measurement tools measure the same construct) [37] of the questionnaire and synthesize the concurrent evidence in children’s populations. Although previous systematic reviews discussed evidences about the convergent validity of PAQ versus accelerometers data reporting low-to-moderate correlation in children [18,22,38], no study specifically focused on the PAQ-C. Therefore, the purpose of this study was to systematically summarize the evidence on the relationship between MVPA measured with an accelerometer device and PAQ-C data. Specifically, we focus our research on MVPA because it is the reference value recommended by the general PA guidelines [1,11]. A systematic review in this field will help to provide a better understanding of the validity of PAQ-C to investigate PA patterns and behaviors in children. In addition, this study will be able to provide evidence-based recommendations for guiding population health programs aiming to promote active living and healthy lifestyles among children.

## 2. Materials and Methods

Four electronic databases including PubMed, Web of Science (WOS), PsycINFO (APAPsycNET), and SCOPUS were considered to search pertinent papers by using search strategies, which were similar and adaptable to each database. Search was performed using the strings (“acceleromet*” OR “motion senso*”) AND (“physical activity questionnair*” OR “PAQ-C”) AND (“child*” OR “young”). The research included the published articles from January 1997 to November 2020. We started from January 1997 because PAQ-C was developed in that period. In addition, references of included articles were screened to identify potential eligible studies.

### 2.1. Eligibility Criteria

The following inclusion criteria were adopted: (1) all participants in the studies were children with an age range 8–14 years; (2) all participants did not present any mental, psychological, physical, or motor disorders; (3) PA was measured objectively using accelerometers and subjectively using the PAQ-C; (4) articles written in English language. No restriction in relation to uniaxial and triaxial devices was performed. Authors were contacted if a study failed to report data about the correlation between accelerometry (i.e., MVPA data) and PAQ-C. Only original, peer-reviewed studies that used the PAQ-C and an accelerometer score were included. If a study presented longitudinal assessment (e.g., physical exercise intervention), we considered only baseline measurements. Other sources (e.g., reviews, meta-analyses, abstracts, opinion articles, books, statements, letters, editorials, comment, and non-peer-reviewed journal articles) were excluded.

### 2.2. Article Selection

All potential studies were imported into Zotero (www.zotero.org, accessed on 21 February 2021), and duplicates were removed. A summary of the study screening protocol and selection has been provided in Figure 1. The selection process was conducted by two authors (D.M. and S.C.), who independently screened the title and/or the abstract of the selected studies to identify studies that potentially met the inclusion criteria. Then, the full texts of the potentially suitable studies were examined by the same authors for eligibility. Disagreements were resolved through discussion with a third author (P.R.B.), who finally decided in case of conflicting results. If any information useful for the data collection was missed, the corresponding author of the manuscript was contacted. If no response was provided or data were not available, the study was excluded from the analysis.

### 2.3. Study Quality Assessment

An adapted version of the Strengthening the Reporting of Observational Studies in Epidemiology checklist (STROBE) [39] was used to assess the quality of the study reporting. The checklist included 20 items grouped into five categories: Abstract (#1 items), Introduction (#2–#3 items), Methods (#4–#11 items), Results (#12–#15 items), and Discussion (#16–#19 items). The assessment of the quality of each item was scored as 1 (i.e., present) or 0 (i.e., not present). The total resultant score (i.e., sum of each item) was considered according to the following level: “low quality” (0–9 points), “medium quality” (10–15 points), and “high quality” (16–19 points). Two independent authors (D.M. and S.C.) completed the study quality assessment. Disagreements were resolved by discussion between assessors.

### 2.4. Meta-Analyses

A Microsoft Excel^®^ (Microsoft Corp., Redmond, WA, USA) spreadsheet was compiled with the following information: study information (i.e., lead author, location, and year of publication); study population characteristics (i.e., sample size, age, and gender); accelerometer information (i.e., placement position, number of days per week and weekend, epoch length, algorithm used to investigate PA intensity, and outcome); principal outcomes (i.e., PAQ-C and MVPA scores) and correlation analysis (i.e., Pearson correlation coefficient or Spearman’s rho). The data were extracted from any section of the manuscript.

The relationship between PAQ-C and MVPA scores was estimated considering the effect size based on the correlation coefficients. Data reported as Spearman’s rho were converted into Pearson correlation coefficient using the formula $r=2sin(r_{s}\frac{\pi }{6}$), where *r* is the Pearson’s correlation coefficient and *r_s_* the Spearman’s rho [40]. Fisher’s transformation was used to convert each correlation coefficient into an approximately normal distribution. The pooled correlations between PAQ-C and MVPA scores were measured by *r* values after converting the Fisher’s z values back into correlation coefficients for presentation. Heterogeneity (i.e., the percentage of the total variability in an effect size between studies) was evaluated using the I^2^ index. The level of heterogeneity represented by the I^2^ index was interpreted as low (25% to ≤50%), moderate (50% to ≤75%), and large (>75%) [41]. A random-effects model was adopted. Publication bias was assessed by visual inspection of the funnel plot and Begg’s and Egger’s tests. All statistical analyses were conducted using the packages “meta” and “metacor” [42] of R (version 4.0.0; R Core Team, Foundation for Statistical Computing, Vienna, Austria). The inverse variance weighting approach was used for pooling [43]. According to Cohen’s guidelines, pooled correlations were interpreted as low (<0.30), moderate (0.31–0.49), and large (≥0.50) [44]. A *p*-value <0.05 was considered statistically significant.

## 3. Results

### 3.1. Studies Systematically Identified

Figure 1 summarizes the systematic search and study selection process.

The initial database search yielded 668 articles. After the removal of the duplicates (N = 301), articles were screened for titles and abstracts, and 23 full-text articles were selected. Five studies were removed because they also included children older than the reference population of the questionnaire, two studies were removed because they were not written in English, and three as they did not report correlation data and the authors did not respond to our request. Overall, 13 studies met the inclusion and reporting criteria.

### 3.2. Study Description

Table 1 summarizes the characteristics of the identified studies according to the following items: general information, sample characteristics, accelerometer information, principal outcomes, and correlation scores between PAQ-C and MVPA scores.

Ten studies were published in the last eight years [16,28,30,31,33,34,35,36,45,46], whereas three studies were published between 1997 and 2011 [27,32,47]. The studies were conducted in Europe [16,28,32,33,34,45], Asia [30,31,35], North America [27,46], North Africa [36], and Oceania [47]. Twelve studies included both male and female participants, whereas one examined girls only [45]. One study [27] did not report data about gender. Sample sizes across studies ranged from 20 to 365; eight studies had sample size ≥100 [28,30,31,32,34,35,45,46]. To objectively investigate PA, 11 studies used ActiGraph [16,28,30,31,32,33,34,35,36,46,47], one the Caltrac [27], and one the Cosmed Liferecorder device [45]. In particular, eight studies adopted triaxial accelerometers [16,28,30,31,33,34,35,36], and five uniaxial accelerometers [27,32,45,46,47].

In all 13 studies included in the meta-analysis, accelerometer devices were worn on the waist. Nevertheless, seven of them placed accelerometers on the right side through an elastic belt [16,31,32,33,34,36,47], whereas the other six studies did not specify on which side the accelerometers were worn [27,28,30,35,45,46]. The accelerometers were worn for seven consecutive days in nine studies [16,27,28,30,31,33,34,35,46], and for five days [32] and four days [47] in one study, respectively. One study [36] did not specify the total days but only four days as the minimum unit of time, whereas another one [45] did not report the days of wearing. The majority of the studies (N = 10) reported an epoch of 15 s or less [16,28,30,31,32,33,34,35,36,45], one of 30 s [46], and the other two did not report the data [27,47].

### 3.3. Study Quality

According to selected criteria, seven studies out of thirteen (54%) were considered “high quality” and six (46%) were considered “medium quality”, while none was considered “low quality”. Criteria commonly absent in reporting were related to defining potential confounders, using adequate power calculations to ensure the study size, and reporting statistical estimate(s) and precision (e.g., 95% CI). Additional information about quality scores is presented in Table 2.

### 3.4. Meta-Analyses

A total of 13 studies were included in the meta-analysis. A random-effects model was adopted. The pooled correlation between PAQ-C and MVPA scores was significant but moderate (r = 0.34, 95% CI [0.29, 0.39], Z = 12.16, *p* < 0.001) and was homogeneous (I^2^ = 24.7%; τ^2^ = 0.0024, *p* = 0.194). Forest plot results are presented in Figure 2.

## 4. Discussion

This systematic review aimed to summarize existing evidence on the convergent validity of the PAQ-C to investigate absolute PA pattern and behavior in children aged 8–14 years. For this purpose, we investigated the aggregated effect size between PAQ-C and accelerometer scores considering 13 publications. The present work is novel because there are only few reviews discussing self-reported questionaries versus objective PA in children [18,22,38], and it focused on a specific questionnaire (i.e., PAQ-C). The PAQ-C is growing in popularity in different sociocultural contexts. Thus, it is necessary to understand whether the PAQ-C may correctly monitor absolute PA in children and accurately assess the effectiveness and changes of interventions designed to increase activity levels, examine relationships between PA and health, and inform public health policy [38]. This aspect is particularly challenging because it may influence healthy lifestyles during adulthood and older age.

A review on the qualitative attributes and measurement properties of PAQ suggested that a correlation coefficient of 0.5 or higher should be acceptable for the validity of PA [48]. The present study disclosed significant moderate pooled correlations of 0.34 [0.29, 0.39] (see Figure 2) with a low degree of heterogeneity among the considered studies (I^2^ = 24.7%, τ^2^ = 0.002, *p* = 0.194). Additionally, our results were quite consistent: none of the selected studies reached the standard of 0.5 in correlation result (range: 0.119–0.456). These data indicate a moderate convergent validity of PAQ-C, although lower than the acceptable standard [48], when compared with the accelerometer. This result suggests a difference in the ability of PAQ-C and accelerometers to measure the same construct (i.e., PA). Thus, it is possible to argue that when investigating absolute PA with PAQ-C, the risk of bias may be large, corroborating the idea of substantial discrepancies between indirect and direct methods to assess PA in pediatric populations [38]. In other words, the wide correlation variability reported in the considered studies indicates that no clear picture can be drawn regarding the PA levels when using PAQ-C.

Our results are in line with previous reviews on children and youth [18,22,38], adult [15,17,37,49], and older population [17] that reported a large variability in correlation results between direct and indirect methods. In an extensive systematic review on the topic in adult population (age ≥ 18 years), Prince at al. [15] found a low-to-moderate correlation (−0.71 to 0.96) between PAQs and accelerometer measures pointing out that the agreement between the two methods was remarkably low. Similarly, Lee et al. [49] reported a greater variability in correlations (from −0.18 to 0.76) between vigorous or moderate activities assessed by means of the International Physical Activity Questionnaire-Short Form and accelerometer measures, as well as an overestimation in PA by 36% to 173% using self-report questionnaires. Again, with a meta-analytic approach similar to that herein presented, Kim et al. [37] examined the convergent validity of the International Physical Activity Questionnaire with an accelerometer and reported an average effect size of 0.21 [0.17, 0.26] with moderate PA. Similar comparisons were conducted in children and youth (5–17 years old). Chinapaw et al. [18] reviewed 61 PAQs for children and adolescents and reported heterogeneity in the results, with correlations between PAQ and accelerometers ranging from very low to high. These data were confirmed in a recent update study that reported lower acceptable validity, partly due to the low methodological quality of the studies [22].

In our analysis, we did not find heterogeneity among the studies (see I^2^ and τ^2^ values), despite the methodological differences of studies included in the meta-analysis. Different types of accelerometers (i.e., uniaxial and triaxial), settings of recorded data (e.g., epoch length or number of days recorded), and data analysis algorithms, such as non-wear-time definition and cutoff point to identify PA intensities (e.g., light, moderate, and vigorous PA), were identified in the selected studies. About two thirds of the studies (i.e., 61.5%) used triaxial rather than uniaxial accelerometers, which are considered more accurate than the latter. Nevertheless, it should be pointed out that all the studies using triaxial accelerometers calculated cutoff PA intensity level through Everson criteria [50], which are based only on the acceleration in the vertical plane. Additionally, we observed a wide variability in epoch lengths setting (ranging from 1 to 30 s). It is well known that the epoch length may affect the estimation of the PA bouts under free-living conditions especially in children [51], and this estimation decreases as epoch length increases [52]. Even if shorter epochs (e.g., 5 s or less) are potentially more sensitive to detect MVPA than longer ones, it seems not to weaken the correlation between PAQ-C and accelerometers. Similarly, while eight studies [16,28,30,31,33,34,35,36] used the international recommend cut-point for children [50,53] to determine the time spent on different PA intensity levels (i.e., range 0–100 counts/min for sedentary behavior, range 101–2295 counts/min for light, range 2296–4011 counts/min for moderate, and ≥4012 counts/min for vigorous PA) five studies [27,32,45,46,47] chose different cut-points. Moreover, while the PAQ-C recalls the PA behavior in the last seven days, 3 out of 13 studies [32,36,45,47] failed to collect the same timeframe due to limiting the accelerometer wearing or considering less than seven days sufficient in the data analysis. Again, not all the studies reported that the self-report and directly assessed PA levels were measured concurrently [30,31,35] leading to an increase of bias. Most of the studies (9 out of 13) [16,27,28,30,31,33,34,35,46] reported requiring that subjects wear the device at least for seven days, but higher variability range in correlation score was reported among these studies (range 0.170–0.390).

All these methodological differences among studies seems not to weaken the correlation between PAQ-C and accelerometers. On the contrary, we were not able to investigate the effect of wearing placement because all included studies placed the accelerometer on the waist. As previously discussed, accelerometers could not measure certain kinds of activities, and the wearing position could amplify this lack. For example, Troiano et al. [54] argued that wearing accelerometers on the wrist could improve wear compliance and allow the measurement of movement during sleep, while Rosemberger et al. [55] claimed that the hip is better for estimating activity energy expenditure and identify activity intensity thresholds. Probably, the appropriate wear location is dependent on specific objectives and will be outlined in the study protocol. For this reason, the effect of wearing placement on correlation between PAQ-C and accelerometers should be taken into account in future studies on the topic.

As a reasonable gold standard for measuring PA does not exist [18], in our study we focused on accelerometers as an objective criterion for measuring PA. Nevertheless, caution is needed when interpreting the findings herein. Indeed, we believe that the results from our study may not be generalized as the overall convergent validity of PAQ-C. Commonly, accelerometer devices are the most used tools to objectively evaluate levels of physical behavior. It allows to directly evaluate the frequency, duration, and intensity of movement and is considered a useful device for estimating energy expenditure [56] in different real contexts. Nevertheless, accelerometer technology did not allow the quantification of some types of activities, such as cycling- and water-based activities or upper body movements, when it is worn on the waist and is not able to discriminate between qualitatively different activities [45,57,58]. On the contrary, the PAQ-C investigates a broad range of leisure and sport activities, which can lead to a larger description of the weekly PA in terms of activity types, time frame during daily and weekly routine, and self-perception of physical activity involvement. Furthermore, differences between PAQ-C score and accelerometers MVPA could be interpreted as the children’s difficulty in judging and feeling their own physical engagement. According to the above findings, our results suggested that PAQ-C and accelerometers may be considered two different tools to investigate PA. In other words, it is possible to suggest that the PAQ-C may be used to investigate children’s PA behaviors (i.e., type and schedule of PA) rather than absolute PA. Concurrently using the two instruments might provide both a qualitative and quantitative evaluation of PA that could permit better exploration of the relation between self-perceived and actual PA in children, to make, for example, corrections to planned activities or to dedicate time to the improvement of self-perception ability. Thus, with the aim of an accurate and effective assessment of PA, it is possible to suggest the use of both instruments to have a deeper overview of PA behavior and level, especially in the first step of the assessment and design of PA interventions [33]. In light of our results, it could be possible to propose the use of the PAQ-C to take information about the type and weekly distribution of physical activities and, thus, provide a global measure of children’s PA, while accelerometers could provide more information about the quantification of the PA.

Some limitations should be finally pointed out. In this systematic review, we focused on the convergent validity of the PAQ-C, but we did not consider measurement properties such as reliability, validity, criterion validity, and measurement error. We considered only correlation coefficients that are not able to detect the agreement and the absolute difference between objective and self-report measures. We focused only on MVPA and not on total PA. It is possible that the correlation between objective and subjective measures of total PA may be larger, due to the lower specificity of the total PA to the use of different cutoffs. Moreover, comparing the PAQ-C score with total PA could overcome difficulties in the perception of physical intensity of children. Additionally, we are not able to investigate gender-related differences in convergent validity. This might be interesting considering the difference in self-perception of PA between males and females [59], even if differences in the considered samples seem not to modify our conclusion, as low heterogeneity shows. Finally, we only focused on English articles disregarding relevant studies published in other languages.

## 5. Conclusions

To our knowledge, the present study is the first comprehensive attempt to synthesize the scientific evidence on the convergent validity of PAQ-C using meta-analysis. Together the results of this study heightened the moderate convergent validity of PAQ-C with accelerometer measurements. Currently, the drawing of any definitive conclusion concerning the convergent validity of PAQ-C in comparison with the accelerometers is not possible. This suggests that the PAQ-C may be valid for identifying children’ PA behavior rather than for absolute PA. Both subjective and objective assessments of PA have limits and advantages. Considering the different nature of both measurements, it is suggested that, for a more comprehensible and deeper PA overview, a combination of PAQ-C and accelerometer devices appears the most promising. Finally, caution should be exerted when comparing studies using PAQ-C and accelerometers.

As practical applications, caution is needed in the choice of instruments to investigate PA in children. PAQ-C and accelerometers seem to measure different aspects of the same construct. While the first one may provide a qualification of PA (e.g., time frame, weekly routine, and self-perception) the second one may provide information about movement quantification (e.g., intensity, frequency, and duration). In addition, the use of both instruments may provide a deeper overview of PA behavior and level, especially in the first step of the assessment and planning of appropriate PA intervention in children. Finally, the use of both instruments allows for a better understanding of the association between actual and perceived physical activity in children.

## Figures and Tables

**Figure 1 ijerph-18-03413-f001:**
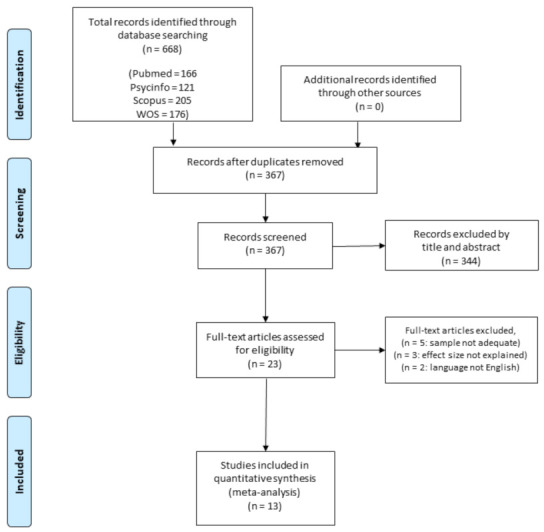
Flow diagram for screening and selection of studies.

**Figure 2 ijerph-18-03413-f002:**
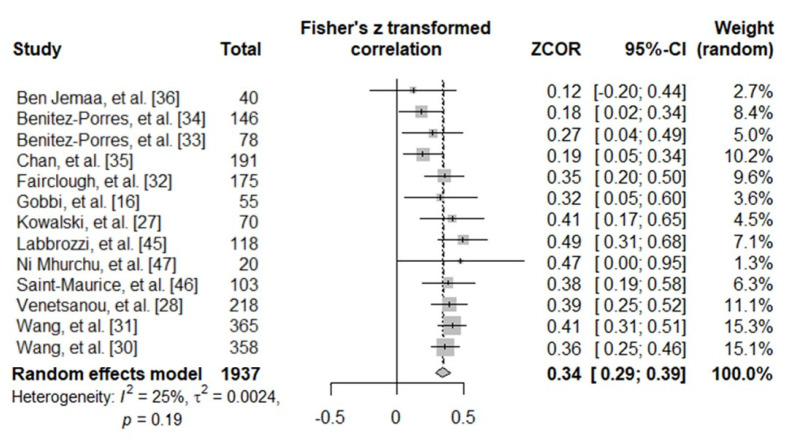
Forest plot showing the relative and pooled correlations between PAQ-C and MVPA scores of the included studies.

**Table 1 ijerph-18-03413-t001:** Summary of the studies included in the meta-analysis (alphabetical order).

Study Information			Study Population				Accelerometer Information						Outcomes		
Authors	Location	Years	Sample Size	Mean Age (Range)	Gender (% Girls)	Model (Axis)	Placement	N Days (Weekend)	Epoch Length (s)	Outcomes	Cut-Point PA Intensity Level (Non-Wearing Definition)	h/Day	PAQ-C (Points)	MVPA (min/Day)	*r*
Ben Jemaa et al. [36]	Tunisia	2018	40	9.34 ± 0.94 (8–11)	47.5%	ActiGraph GT3X + (triaxial)	hip	4 (1)	15	ST, LPA, MPA VPA, MVPA	Evenson et al. (≥60 min)	≥6	2.55 ± 0.67	59.77 ± 22.01	0.119
Benitez-Porres et al. [34]	Spain	2016	146	10.8 ± 1.3 (9–12)	43.1%	ActiGraph GT3X (triaxial)	hip	7 (1)	1	MVPA step/day	Evenson et al. (≥60 min)	≥10 (week) ≥8 (WE)	3.09 ± 0.64	62.80 ± 13.90	0.170 ^¥^
Benitez-Porres et al. [33]	Spain	2016	78	10.98 ± 1.17 (9–12)	46.1%	ActiGraph GT3X (triaxial)	hip	7 (1)	1	MVPA	Evenson et al. (≥60 min)	≥10 (week) ≥8 (WE)	3.24 ± 0.64	63.22 ± 14.40	0.248 ^¥^
Chan et al. [35]	China	2018	191	9.9 ± 1.0 (8–11)	59,7%	ActiGraph GT3X + (triaxial)	hip	7 (1)	15	MVPA	Evenson et al. (≥20 min)	≥6	2.67 ± 0.70	40.86 ± 14.07	0.190
Fairclough et al. [32]	England	2011	175	10.6 ± 0.3 (10–11)	55.4%	ActiGraph GT1M (uniaxial)	hip	5 (1)	5	MPA, VPA, MVPA, counts/min	Ekelund et al. (≥20 min)	≥6 (week) ≥6 (WE)	3.39 ± 0.13 (M) 3.00 ± 0.11 (F)	66.30 ± 3.70 (M) 54.10 ± 3.20 (F)	0.338 ^φ^
Gobbi et al. [16]	Italy	2016	55	9.5 ± 0.4 (9–10)	50.9%	ActiGraph GT3X + (triaxial)	hip	7 (n.r.)	15	MVPA	Evenson et al. (≥60 min)	≥9	2.79 ± 0.52	n.r.	0.300 ^¥^
Kowalski et al. [27]	Canada	1997	70	11.30 ± 1.39 (9–13)	n.r.	Caltrac (uniaxial)	hip	7 (1)	n.r.	MVPAMVPA > 10min	n.r.(n.r.)	n.r.	3.32 ± 0.68	n.r.	0.390
Labbrozzi et al. [45]	Italy	2012	118	n.r. (11–13)	100%	COSMED Lifecorder (uniaxial)	hip	n.r.	4	LPA, MPA, VPA	Kumahara et al. (n.r.)	n.r.	n.r.	n.r.	0.456 ^φ^
Ni Mhurchu et al. [47]	New Zealand	2008	20	12 ± 1.5 (10–14)	40%	ActiGraph 7164 (uniaxial)	hip	4 (2)	n.r.	PA counts, LPA, MPA, VPA	Freedson et al. (≥20 min)	≥8	1.8 ± 0.6	n.r.	0.440 ^φ^
Saint-Maurice et al. [46]	USA	2014	103	10.8 ± 2.0 (8–13)	52.4%	ActiGraph GT1M (uniaxial)	hip	7 (1)	30	MVPA	Freedson et al. (≥90 min)	≥9	3.1 ± 0.7	n.r.	0.350
Venetsanou et al. [28]	Greece	2020	218	10.99 ± 1.52 (9–13)	56.9%	ActiGraph GT3X + (triaxial)	hip	7 (1)	5	MVPA, steps/day	Evenson et al. (n.r)	n.r.	2.70 ± 0.55 (M) ^φ^ 2.51 ± 0.53 (F)	42.46 ± 12.46 (M) ^φ^ 31.70 ± 9.21 (F)	0.354 ^¥^
													2.78 ± 0.37 (M) 2.35 ± 0.47 (F)	40.33 ± 11.95 (M) 33.31 ± 8.41 (F)	
Wang et al. [31]	China	2016	365	10.2 ± 1.1 (8–13)	45.2%	ActiGraph GT3X (triaxial)	hip	7 (1)	5	MVPA	Evenson et al. (≥20 min)	≥8	2.70 ± 0.70	43.10 ± 12.74	0.390
Wang et al. [30]	China	2016	358	10.5 ± 1.1 (9–12)	45.8%	ActiGraph GT3X (triaxial)	hip	7 (1)	5	MPA, VPA, MVPA	Evenson et al. (≥20 min)	≥8	2.60 ± 0.68	43.00 ± 13.72	0.330 ^¥^

Notes: n.r., data not reported in the paper; M, male; F, female; SB, Sedentary Behavior; LPA Light Physical Activity; MPA, Moderate Physical Activity; VPA, Vigorous Physical Activity; MVPA, Moderate to Vigorous Physical Activity; WE, weekend; ^¥^, data are reported as Spearman’s rho; ^φ^, data directly provided by the authors. Cut-point PA intensity level: Evenson’s PA cutoff: SB (0–100 counts/min), LPA (101–2295 counts/min), MPA (2296–4011 counts/min), VPA (≥4012 counts/min); Ekelund’s PA cutoff: SB (<500 counts/min), LPA (501–2000 counts/min), MPA (2001–3999 counts/min), VPA (≥4000 counts/min); Freedson’s PA cutoff: LPA (1.5–2.9 MET), MPA (3.0–5.9 MET), VPA (≥ MET); Kumahara’s PA cutoff: LPA (<3 MET), MPA (3–6 MET), VPA (≥6 MET). PAQ-C, Physical Activity Questionnaire for Children.

**Table 2 ijerph-18-03413-t002:** The Strengthening the Reporting of Observational Studies in Epidemiology checklist (STROBE) scores and summary of studies’ quality.

	1#	2#	3#	4#	5#	6#	7#	8#	9#	10#	11#	12#	13#	14#	15#	16#	17#	18#	19#	Score/19
Ben Jemaa et al. [36]	1	1	1	1	1	0	1	1	0	1	1	0	1	1	0	1	1	1	0	14
Benitez-Porres et al. [34]	1	1	1	1	1	0	1	1	0	1	1	1	1	1	0	1	1	1	0	15
Benitez-Porres et al. [33]	1	1	1	1	1	0	1	1	1	1	1	1	1	1	1	1	1	1	0	17
Chan et al. [35]	1	1	1	1	1	1	0	1	0	1	1	1	1	1	0	1	1	1	1	16
Fairclough et al. [32]	1	1	1	1	0	1	1	1	0	1	1	1	1	1	0	1	1	1	1	16
Gobbi et al. [16]	1	1	1	1	0	0	1	1	0	1	1	1	1	1	0	1	0	1	1	15
Kowalski et al. [27]	1	1	1	1	1	0	0	0	0	0	0	1	1	1	0	1	1	1	1	12
Labbrozzi et al. [45]	1	1	1	1	1	1	1	1	0	1	1	0	1	1	0	1	1	1	1	16
Ni Mhurchu et al. [47]	1	1	1	1	1	1	1	0	0	1	1	1	1	1	1	1	0	0	0	14
Saint-Maurice et al. [46]	1	1	1	1	0	0	1	1	0	1	1	0	1	1	0	1	1	1	1	14
Venetsanou et al. [28]	1	1	1	1	1	1	1	1	0	1	1	1	1	1	0	1	0	1	1	16
Wang et al. [31]	1	1	1	1	1	0	1	1	0	1	1	1	1	1	0	1	1	1	1	16
Wang et al. [30]	1	1	1	1	0	1	1	1	0	1	1	1	1	1	0	1	1	1	1	16

Notes: 0 = Item criterion is absent or insufficient information is provided; 1 = item criterion is present and explicitly described. #1. In the abstract, an informative and balanced summary of what was done and what was found is provided. #2. Explains the scientific background and rationale for the investigation being reported. #3. States clear, specific objectives and/or any prespecified hypotheses. #4. Describes the setting (e.g., school context), locations (e.g., nation), and relevant dates for data collection. #5. Give characteristics of study participants (must include age and gender) and eligibility criteria. #6. Clearly defines all outcomes, potential confounders, and effect modifiers. #7. For each variable of interest, gives sources of data and details of methods of assessment (e.g., information about accelerometer time of wearing, epoch length, wearing position). #8. Describes any efforts to address potential sources of bias (e.g., minimum of daily wearing, statistical treatment of outliers). #9. Checks whether the study used power calculations to ensure the study size was adequately powered to detect hypothesized relationships? #10. Explains how quantitative variables were handled in the analyses. If applicable, describe which groupings were chosen, and why. #11. Describes all statistical methods, including those used to control for confounding and any methods used to examine subgroups and interactions (if applicable). #12. Indicates the number of participants with missing data for each variable of interest. #13. Cohort study—Report numbers of outcome events or summary measures over time. #13. Cross-sectional study—Reports numbers of outcome events or summary measures. #14. A measure of effect size is provided (e.g., Cohen’s effect size, Pearson’s r, Spearman’s rho). #15 Provides statistical estimate(s) and precision (e.g., 95% CI) for each sample or subgroup group examined. #16. A summary of key results with reference to study objectives is provided. #17. Discusses limitations of the study, considering sources of potential bias, confounding factors, or imprecision. #18. A cautious overall interpretation of results considering objectives and relevant evidence. #19. Discusses the generalizability of the study results to similar or other contexts. TOTAL/19.

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
