# Peer review of "Subjective versus Objective Measure of Physical Activity: A Systematic Review and Meta-Analysis of the Convergent Validity of the Physical Activity Questionnaire for Children (PAQ-C)"

_ijerph, 2021, doi:10.3390/ijerph18073413_

Round 1
Reviewer 1 Report
Overall speaking, the writing of the paper was clear and the study design was sound. Nonetheless, I have several questions/comments on certain areas of the paper, and I hope these could be considered by the authors:
Line 23-25, method, results:
From my understanding, the choice between fixed vs random effects model is a decision that should be made a prior, and not a decision based on the I-squared. If it was decided that a fixed effect model should be used, the I-squared shouldn't even be calculated in the first place. Thus I think the more appropriate way is to say the random effect model was adopted, only that no heterogeneity was detected.
Line 24:
What does "marginally" moderate mean?
Line 26, discussion:
"heightened" a moderate relationship? I'm a bit confused what this meant. Did the authors mean "highlighted" instead?
Line 27:
I don't think any study could give you a "definite conclusion", so this sentence seems a bit redundant. In fact, I would say that the results provide fairly strong evidence that the correlation between PAQ-C and accelerometer measures is around .30-.40.
Line 51:
"heart rate variability" or "heart rate"? I thought heart rates are more commonly used as a measure of PA instead of HRV? (please correct me if I'm wrong)
Line 60:
"measure forces produced" - they measure acceleration, not force.
Line 66-78:
I think it would be good to mention what other questionnaires are available and how they differ from the PAQ-C.
Line 79, discussion:
I am not sure if the "growth in popularity" statement is justified by only a few studies using the questionnaire. I personally feel that's an overstatement.
Line 99:
the decision to start from 1997 is not "arbitrary" at all, since a justification was provided.
Line 108:
when is "when necessary"? please elaborate.
Line 141:
what is "sen" in the conversion formula? I'm unfamiliar with this expression and I suspect a few other readers would have the same question.
Line 164:
Could the authors elaborate on why studies were removed due to an inadequate sample? An inclusion/exclusion criterion based on sample size was not mentioned beforehand. So removing studies based on sample size alone does not seem warranted.
Line 195:
For the sake of this meta-analysis, how would a lack of power calculation or provision of 95% CI's lead to potential biases that would thwart the accuracy of the results of the meta-analyses?
Line 254:
ActiGraph devices are triaxial but the most common cutoff (Evenson) only uses the acceleration on one axis. So effectively the results derived are uniaxial in nature as well.
Line 259:
I don't think the authors could claim that the other factors do not influence the correlation. So make that claim, you'd have to compare the correlations using the same dataset but by exacting the data using different criteria/cutpoints. It's probably fairer just by saying the findings are robust to these other decisions.
Line 312:
The PAQ-C was never intended to measure an activity level that is comparable to the existing MVPA guidelines (i.e., you can't estimate the MVPA/day based on questionnaire responses), so obviously we can't use it to test if it over- or underestimates MVPA. I felt this is a rather important distinction between these two measures of PA, and probably should have been mentioned in the introduction as well.
Line 337:
I don't really see how the use of both measures could lead to better interventions for the reduction of sedentary behaviors, since PAQ-C doesn't actually measure SB. I suggest the authors elaborate on this point, or simply remove this argument.
Author Response
Overall speaking, the writing of the paper was clear and the study design was sound. Nonetheless, I have several questions/comments on certain areas of the paper, and I hope these could be considered by the authors:
Authors: We really appreciated the constructive feedback provided by the reviewer as well as his/her effort in making our manuscript clearer and more readable. Thank you.
Line 23-25, method, results:
From my understanding, the choice between fixed vs random effects model is a decision that should be made a prior, and not a decision based on the I-squared. If it was decided that a fixed effect model should be used, the I-squared shouldn't even be calculated in the first place. Thus I think the more appropriate way is to say the random effect model was adopted, only that no heterogeneity was detected.
Authors: We thank for the suggestion and are agree with this comment. Therefore, also according to a previous methodological study (e.g., Tufanaru et al., 2015), we decided to perform our meta-analysis starting from a random-effects model. As a consequence, we changed the abstract, the statistical analysis, and results sections according to this comment.
Line 24:
What does "marginally" moderate mean?
Authors: The term “marginally’ may be confounding. Therefore, we deleted it and rephased the sentence (“The pooled correlation between PAQ-C and MVPA scores was significant but with a moderate effect size (r=0.34 [0.29, 0.39], Z=15.00, p<0.001).”)
Line 26, discussion:
"heightened" a moderate relationship? I'm a bit confused what this meant. Did the authors mean "highlighted" instead?
Authors: We made a type error. Therefore, we reported the right word (“highlighted”). In addition, according to this comment, we have grammatically reviewed the entire manuscript.
Line 27:
I don't think any study could give you a "definite conclusion", so this sentence seems a bit redundant. In fact, I would say that the results provide fairly strong evidence that the correlation between PAQ-C and accelerometer measures is around .30-.40.
Authors: We thank for the suggestion and deleted the sentence.
Line 51:
"heart rate variability" or "heart rate"? I thought heart rates are more commonly used as a measure of PA instead of HRV? (please correct me if I'm wrong)
Authors: We thank the reviewer for the suggestion. We agree with the reviewer’s point of view and deleted the term “variability”.
Line 60:
"measure forces produced" - they measure acceleration, not force.
Authors: We thank the reviewer for the suggestion. We agree with the reviewer’s point of view and changed the word.
Line 66-78:
I think it would be good to mention what other questionnaires are available and how they differ from the PAQ-C.
Authors: We thank the reviewer for his/her suggestion. Therefore, we decided to cite the review of Hidding et al. (2018) where readers could find all the questionnaires available as well as the differences among them. However, we would like to maintain the flow of the introduction avoiding of opening a new paragraph about PA questionnaires for children, which could risk leading readers out of the focus (i.e., main topic/aim) of the present study.
Line 79, discussion:
I am not sure if the "growth in popularity" statement is justified by only a few studies using the questionnaire. I personally feel that's an overstatement.
Authors: We agree with the reviewer’s point of view and decided to rephrase the sentence according to the suggestion.
Line 99:
the decision to start from 1997 is not "arbitrary" at all, since a justification was provided.
Authors: Thank you. We corrected according to this comment.
Line 108:
when is "when necessary"? please elaborate.
Authors: We elaborated the sentence as requested.
Line 141:
what is "sen" in the conversion formula? I'm unfamiliar with this expression and I suspect a few other readers would have the same question.
Authors: We are sorry for the type error. The term “sen” was changed in “sin”, which is the expression indicating the mathematical function sine.
Line 164:
Could the authors elaborate on why studies were removed due to an inadequate sample? An inclusion/exclusion criterion based on sample size was not mentioned beforehand. So removing studies based on sample size alone does not seem warranted.
Authors: Thanking for having highlighted this point, we have specified this part of the “Result” section as follows:
“Five studies were removed because included also older children than the reference population of the questionnaire, two studies were removed because they were not written in English, and three as they did not report correlation data and authors did not respond to our request.”
Line 195:
For the sake of this meta-analysis, how would a lack of power calculation or provision of 95% CI's lead to potential biases that would thwart the accuracy of the results of the meta-analyses?
Authors: We agree with your argument, the lack in the methodological approach of most of researches on PAQ is a problem already highlighted in other studies. Therefore, we underlined this issue in the conclusion section.
Line 254:
ActiGraph devices are triaxial but the most common cutoff (Evenson) only uses the acceleration on one axis. So effectively the results derived are uniaxial in nature as well.
Authors: Thank you to highlight this point. We added this aspect in this part of the manuscript.
Line 259:
I don't think the authors could claim that the other factors do not influence the correlation. So make that claim, you'd have to compare the correlations using the same dataset but by exacting the data using different criteria/cutpoints. It's probably fairer just by saying the findings are robust to these other decisions.
Authors: Thanking for the suggestion, we moderated our speculation.
Line 312:
The PAQ-C was never intended to measure an activity level that is comparable to the existing MVPA guidelines (i.e., you can't estimate the MVPA/day based on questionnaire responses), so obviously we can't use it to test if it over- or underestimates MVPA. I felt this is a rather important distinction between these two measures of PA, and probably should have been mentioned in the introduction as well.
Authors: We agree with the argument. At line 312 we don’t refer to the level of MVPA, but to the total PA. Nevertheless, we realize that this sentence may be misunderstood by the readers. Thus, we decided to delete the sentence. In addition, we specified in the introduction section the aim of the PAQ-C.
Line 337:
I don't really see how the use of both measures could lead to better interventions for the reduction of sedentary behaviors, since PAQ-C doesn't actually measure SB. I suggest the authors elaborate on this point, or simply remove this argument.
Authors: Thanking for this suggestion, we deleted the part including the SB.
Reviewer 2 Report
This systematic review and meta-analysis review aiming to analyse the relationship between Moderate and Vigorous Physical 14 Activity (MVPA) provided by accelerometer devices and Physical Activity Questionnaire for 15 Children (PAQ-C). This paper tries to explain in a positive way when looking at the topic of health in young populations and methodologies to assess physical activity. I think some things need clarifying for the publication that will help in the overall interpretation and understanding of the results before to be published within the scope of IJERPH.
Introduction
Comment 1: The authors have presented a clear and well-structured introduction.
Comment 2: (Page 2 – lines 63-65) Please add a reference.
Comment 3: (Page 2 – lines 66-68) The PAQ-C were developed to assess general levels of physical activity, not only MVPA. To avoid misconceptions, I suggest to the authors, rewrite this phrase including “...general levels of physical activity.”. (The Physical Activity Questionnaire for Older Children (PAQ-C) and Adolescents (PAQ-A) Manual)
Comment 4: (Page 2 – lines 66-68) Are the PAQ-C appropriate used in summer or holiday periods? Can the authors be more specific? The authors should cite this specificity.
Results
Comment 5: (Page 4 – line 158) Figure 1 - To better understand the picture, add the total number of records.
Comment 6: (Page 4 – line 158) Figure 1 - Why these records were excluded? By title, abstract? Please add information.
Comment 7: (Page 6 – line 171) Table 1 - Legend: “…MPA (2001….) Please add a comma.
Discussion
Comment 8: (Page ? – line 288-293): Please add references.
Comment 9: (Page ? – line 288-293): In table 1 were observed several brands and models of accelerometers, as well as different cut-points of physical activity intensity to identify the MVPA. These facts can influence the correlation analysis between objective and subjective measures. What are the authors think about this fact?
Author Response
This systematic review and meta-analysis review aiming to analyse the relationship between Moderate and Vigorous Physical 14 Activity (MVPA) provided by accelerometer devices and Physical Activity Questionnaire for 15 Children (PAQ-C). This paper tries to explain in a positive way when looking at the topic of health in young populations and methodologies to assess physical activity. I think some things need clarifying for the publication that will help in the overall interpretation and understanding of the results before to be published within the scope of IJERPH.
Introduction
Comment 1: The authors have presented a clear and well-structured introduction.
Authors: We thank the reviewer for his/her appreciation about the Introduction section.
Comment 2: (Page 2 – lines 63-65) Please add a reference.
Authors: We added the reference as requested.
Comment 3: (Page 2 – lines 66-68) The PAQ-C were developed to assess general levels of physical activity, not only MVPA. To avoid misconceptions, I suggest to the authors, rewrite this phrase including “...general levels of physical activity.”. (The Physical Activity Questionnaire for Older Children (PAQ-C) and Adolescents (PAQ-A) Manual).
Authors: We thank the review for this suggestion and coherently changed the sentence.
Comment 4: (Page 2 – lines 66-68) Are the PAQ-C appropriate used in summer or holiday periods? Can the authors be more specific? The authors should cite this specificity.
Authors: Thanks for the suggestion, we specified this point.
Results
Comment 5: (Page 4 – line 158) Figure 1 - To better understand the picture, add the total number of records.
Authors: We added the total number of records according to the suggestion.
Comment 6: (Page 4 – line 158) Figure 1 - Why these records were excluded? By title, abstract? Please add information.
Authors: We specified the reason of exclusion as suggested.
Comment 7: (Page 6 – line 171) Table 1 - Legend: “…MPA (2001….) Please add a comma.
Authors: We added the comma as suggested.
Discussion
Comment 8: (Page ? – line 288-293): Please add references.
Authors: We added the reference as requested.
Comment 9: (Page ? – line 288-293): In table 1 were observed several brands and models of accelerometers, as well as different cut-points of physical activity intensity to identify the MVPA. These facts can influence the correlation analysis between objective and subjective measures. What are the authors think about this fact?
Authors: We discussed the effect of different characteristics of devices and methods of analysis (i.e. cut-point, epoch, etc.) in the discussion section. Our results show that these differences seem not to weaken the correlation between PAQ-C and accelerometers.
Round 2
Reviewer 1 Report
I would like the authors for addressing the comments I made previously. I have reviewed the revised version of the paper and have some minor comments/ suggestions as listed below. Also, some minor grammatical issues are still present, so I urge the authors to review the language used in the paper in detail again.
Line 26-28:
"In conclusion, the results highlighted a moderate relationship (around 0.30-0.40) between PAQ-C and accelerometer measurements and suggested to concurrently consider the results of both tools for a more comprehensible description, in terms of quality and quantity."
This sentence doesn't read very well. Perhaps it's better to break it down into two separate sentences.
Line 49-50:
"(e.g., accelerometer and pedometers devices and heart rate)"
Replace the first "and" by a comma?
Line 56:
"correctly understand questions" is a strange expression to me. Delete "correctly" or perhaps "correctly interpret questions" might be better?
Line 59-61:
"Accelerometer technology allows to measure accelerations produced by movement and may be considered an affordable and feasible instrument to measure PA levels."
I think many researchers would tell you they aren't that affordable at all!
Line 153:
I suggest adding a sentence here to describe the meta-analytical approach being used. I believe the R package uses the Borenstein et al (inverse variance as weighing) approach for doing meta-analyses? Adding a sentence here would clarify that as some readers would like to know.
Line 206:
"lowly heterogeneous" - Perhaps just say it was homogeneous?
Line 253-256:
I am still not convinced that the lack of power analyses in primary studies would have a significant impact on the results of the meta-analyses. If the sample size was small, their weighing in the meta-analyses would also be small. If they affect the results, you could probably expect to see heterogeneity in the effect size, but this wasn't the case.
Thus my suggestion is to remove these two sentences completely. Nonetheless, if the authors feel strongly that this should be kept, I'd be willing to agree to disagree on this.
